# Cervical Cell/Clumps Detection in Cytology Images Using Transfer Learning

**DOI:** 10.3390/diagnostics12102477

**Published:** 2022-10-13

**Authors:** Chuanyun Xu, Mengwei Li, Gang Li, Yang Zhang, Chengjie Sun, Nanlan Bai

**Affiliations:** 1School of Artificial Intelligence, Chongqing University of Technology, Chongqing 400054, China; 2College of Computer and Information Science, Chongqing Normal University, Chongqing 401331, China

**Keywords:** cervical cancer, transfer learning, faster R-CNN, multi-scale training, bounding box loss

## Abstract

Cervical cancer is one of the most common and deadliest cancers among women and poses a serious health risk. Automated screening and diagnosis of cervical cancer will help improve the accuracy of cervical cell screening. In recent years, there have been many studies conducted using deep learning methods for automatic cervical cancer screening and diagnosis. Deep-learning-based Convolutional Neural Network (CNN) models require large amounts of data for training, but large cervical cell datasets with annotations are difficult to obtain. Some studies have used transfer learning approaches to handle this problem. However, such studies used the same transfer learning method that is the backbone network initialization by the ImageNet pre-trained model in two different types of tasks, the detection and classification of cervical cell/clumps. Considering the differences between detection and classification tasks, this study proposes the use of COCO pre-trained models when using deep learning methods for cervical cell/clumps detection tasks to better handle limited data set problem at training time. To further improve the model detection performance, based on transfer learning, we conducted multi-scale training according to the actual situation of the dataset. Considering the effect of bounding box loss on the precision of cervical cell/clumps detection, we analyzed the effects of different bounding box losses on the detection performance of the model and demonstrated that using a loss function consistent with the type of pre-trained model can help improve the model performance. We analyzed the effect of mean and std of different datasets on the performance of the model. It was demonstrated that the detection performance was optimal when using the mean and std of the cervical cell dataset used in the current study. Ultimately, based on backbone Resnet50, the mean Average Precision (mAP) of the network model is 61.6% and Average Recall (AR) is 87.7%. Compared to the current values of 48.8% and 64.0% in the used dataset, the model detection performance is significantly improved by 12.8% and 23.7%, respectively.

## 1. Introduction

Cervical cancer is the fourth most common cancer and also the second most common disease among women living in developing and low-income countries [1]. According to WHO statistics, in 2018, there were about 570,000 new cases and about 311,000 women died from this fatal disease worldwide [2]. More than 80% of cervical cancer cases and 85% of deaths occur in low-income and developing countries due to a lack of screening and treatment facilities [3]. Studies have shown that human papillomavirus (HPV) is a major cause of cervical cancer [4], and in recent years, HPV-mediated cervical disease has experienced a steep decline in some countries with the intervention of HPV vaccines. However, widespread implementation has been limited by economic factors, different health care priorities, and considerations of vaccine availability in different countries [5]. Cervical cancer is a type of curable cancer, with a cure rate of 92% if cervical cancer lesions are detected in the early stages and treated adequately [6]. If women worldwide were screened for cervical cancer at an early stage every five years, the mortality rate from cervical cancer would likely be reduced by 60% [7]. Therefore, to reduce the threat of cervical cancer to women’s health, it is very important that early screening is performed in addition to active prevention. Cervical cancer screening is a routine screening in women’s health [8], and TCT (thin-layer liquid-based cytology) is the most commonly used type of cytologic screening. In the medical field, the diagnosis of cervical cancer is mainly based on the pathological morphology in cervical cytopathological images to determine whether the disease is present. This method requires the human eye to observe samples through a microscope, which requires high expertise and experience. It typically takes 10 years to cultivate talent in this field [9]. In addition, the analysis of pathological images requires considerable time, is somewhat subjective, and is a tedious and error-prone task. Beginning in the 1980s, computer-aided diagnosis (CAD) systems were introduced to help physicians interpret medical images and improve their efficiency [10].

In recent years, with deep learning performing well in many tasks, such as [11,12,13], etc., many researchers have used deep learning methods in the development of automated assisted screening methods for cervical cancer. Most of these studies focus on cell classification, such as those in [14,15,16,17,18,19,20,21,22,23,24,25]. Large datasets are very important for high performance deep convolutional networks, considering the limited cervical cell annotation data (e.g., the Herlev benchmark dataset [26] has only 917 cells, with 675 abnormal cells and 248 normal cells out of 917 cells). Some studies have used transfer learning [27] to address this problem. There are also studies that used deep learning detection algorithms to identify and localize cervical cells directly [28,29,30,31,32,33,34,35,36,37,38,39,40,41]. In such studies, the datasets used were generally private and not publicly available. The amount of data and annotations in these datasets is generally not very large, and some of these studies use transfer learning methods to improve network model performance. In cervical cell detection methods that use transfer learning [29,30,32,35,36], the weights are initialized in the same way as those used in cervical cell classification studies, which employ transfer learning methods, i.e., the network parameters are initialized using a model pre-trained on the classification problem via ImageNet [42]. However, detection and classification are two different tasks. Considering this factor, our study proposes using the COCO [43] pre-trained model to initialize network parameters when using deep learning detection algorithms for cervical cell detection, and performing fine-tuning accordingly to obtain the best performance of the network model. This study conducted extensive experiments to demonstrate that network model initialization using COCO [43] pre-trained models was better than using ImageNet [42] pre-trained models in the cervical cell detection study.

In the study of classification tasks, the processing aspect of the input image is usually resized to one size (e.g., classification with ImageNet [42], where the input image is processed to 224 × 224). However, the detection task differs from the classification task in that the detection task has objects of different scales in the input images. To improve the robustness of the network model to cervical cells of different scales, In this study, multi-scale training is carried out according to the actual situation of the dataset and transfer learning.

Bounding box loss is very important for the accurate localization of object detection, and the optimization history of bounding box loss in recent years: L1, L2 loss; SmoothL1 loss; IoU loss [44]; GIoU loss [45]; DIoU loss [46]; CIoU loss [46]. L1 loss and L2 loss assume the bounding box as four independent variables for optimization, and SmoothL1 loss combines the advantages of L1 loss and L2 loss. IoU (Intersection over Union) loss regresses the prediction box as a whole, GIoU (Generalized IoU) solves the disadvantages of IoU while making full use of its advantages. DIoU (Distance IoU) loss takes into account both the overlapping area of the bounding box and the distance of the center point, which converges faster. CIoU (Complete IoU) loss summarizes three important geometric measures that a good bounding box regression loss should consider, i.e., overlap area, center point distance, and aspect ratio. This loss function can lead to faster convergence and better performance. Considering the impact of bounding box loss on the detection precision of cervical cells/clumps, we analyze the impacts of different bounding box loss functions on model performance based on transfer learning. Finally, we select the most suitable bounding box loss function (SmoothL1) for the current practical situation.

Our contributions can be summarized as follows: (1) We propose using COCO pre-trained model weights to initialize the network model when using deep learning algorithms for cervical cancer cell detection and use the fine-tuning method of transfer learning to obtain the best performance, thereby better addressing the problem of the limited dataset at training time; (2) we performed multi-scale training to improve the detection performance of the network model for Cervical cells/clumps based on the pre-trained model and the actual situation of the dataset; (3) we achieved the highest mAP (61.6%) and AR (87.7%) on the current cervical cell dataset from [35].

## 2. Materials and Methods

In this study, the backbone of the network model is ResNet [47], with the Faster R-CNN [48] + FPN [49] detection algorithm used as a benchmark. Transfer learning is carried out using the COCO pre-trained model for fine-tuning. The training method uses multi-scale training. The overall network structure of the model is shown in Figure 1.

### 2.1. CNN-Based Object Detection

CNN-based object detection algorithms can be mainly divided into two categories: One is two-stage detection algorithms, including classical algorithms such as R-CNN [50], Fast R-CNN [51], and Faster R-CNN [48]. Another class includes one-stage detection algorithms, such as SSD [52], YOLO series [53,54,55], RetinaNet [56], and FCOS [57]. Two-stage detection algorithms have higher precision for localization and target identification, whereas one-stage detection algorithms have a higher inference speed. The first stage of the two-stage detection algorithm, the RPN (Region Proposal Network) stage, perform region suggestion to select ROIs (Regions of Interest) and extracts image features, whereas the second stage intercepts features for each ROI region via ROI Pooling (or ROI Align) from the feature map and converts them to the same size feature output for the next category-specific classification and precise bounding box localization operations. One-stage algorithms have no region suggestion step and operates directly on the input image to predict the class and bounding box of the object. Here, we use Faster R-CNN [48] + FPN [49] as a baseline. The first reason for this choice is that it is still very advanced and offers considerable flexibility. The second reason is that the dataset used in this study is from [35], thus, we selected the same baseline as [35] to improve the validation and comparison of the effectiveness of the methods used in this study.

### 2.2. Transfer Learning

Transfer learning is a subfield of deep learning that focuses on transferring knowledge from source data to a target domain to enhance the target task [58]. By transferring knowledge from large public datasets (e.g., ImageNet [42]) to domain-specific tasks (e.g., cervical cell classification task [14,16]), the problem of overfitting can be reduced, the generalization of the model can be improved, and the efficiency of model training can be enhanced. Fine-tuning is a transfer learning method that is commonly used as an effective training strategy in various deep learning tasks. The following three ways are commonly used: After the network model is initialized with pre-trained weights. The first way is train current network model directly. The second is to freeze all the convolutional layers of the network model and train only the fully connected layers that are adapted to target tasks. The third is to freeze part of the convolutional layers of the network model (usually the convolutional layers near the input part) and train the remaining convolutional and fully connected layers. The first way is usually used when the target task data volume is large and has high similarity to the source data. The second way is used when the target task data volume is small, but the similarity with the source data is very high. The dataset used in this study includes 6666 images in the training set, which is not a large amount of data. Additionally, the dataset is of a cell type, which is not very similar to natural image datasets such as ImageNet [42] and the COCO dataset [43]. Therefore, according to the actual situation, we selected the third way to fine-tune the network model. The study performed Fine-tuning by freezing the first few stages in the backbone (ResNet [47]) or freezing the stem and other stages. Figure 2 shows the network freezing model of the study. The model was fine-tuned for better feature learning by freezing some convolutional layers during the training stage of the network.

### 2.3. Multi-Scale Training

The multi-scale problem of object detection is a matter of concern [59]. In response, researchers have explored, in detail, how to accurately detect objects of different scale sizes in the input image, such as FPN [49] which fuses deep and shallow features on feature maps of different scales. This method is used for image feature processing. However, the size of the input image also has a great impact on the performance of the detection model. In terms of processing the input image, a multi-scale approach can be used, i.e., the input image is no longer processed into one scale as in the classification problem. Similar to image pyramids [60], but unlike image pyramids, instead of scaling the input image down to multiple scales and computing the feature maps separately for each scale and performing subsequent detection, multi-scale training sets up multiple different scales to choose from, and at each iteration, one scale is randomly selected from among the multiple scales to process the input image. Although only one size is used each time, that size is different each time. Under this approach, the robustness of the detection model to object size is increased. The advantage of this training approach is that the model training time is much less than that when using image pyramids, but the model detection performance is still very good. The present study analyzes the actual situation of the used dataset and calculates the average scale of the training set images, i.e., mean_images_width: 1311.1 px and mean_images_height: 724.6 px. The multi-scale we chose is [(1333,640), (1333,800)]. There are two reasons for using this scale: One is that this is the same scale used in multi-scale training to obtain the COCO pre-trained model. Here, we use this scale to better utilize the COCO multi-scale pre-trained model. Second, this multi-scale is just around the average scale of the dataset, which is more in line with the actual situation of the dataset. Subsequent experiments also confirmed that using this multi-scale works well. For each iteration of the input image, a scale is randomly selected from the two scales to resize the image (width: w, height: h), and the original image ratio is maintained when resizing, as summarized in Table 1. 

### 2.4. Bounding Box Loss

Bounding box regression is the key step in object detection. The “n-paradigm” (e.g., L1 loss and L2 loss) is widely used in bounding box regression, where the object bounds are considered as four independent variables and the four values of the object bounds are optimized, but the evaluation metric is the IoU (Intersection over Union) metric. The “n-paradigm” is not suitable, however, for the evaluation metric. To solve this problem, IoU loss was proposed in [44] to predict the bounding box, which regresses the four bounds of the predicted box as a whole instead of the four independent variables. This loss function speeds up the convergence of the model and improves the localization precision. IoU loss has some weaknesses, however. One weakness is that when the predicted box and object box do not intersect, IoU = 0, which cannot reflect the distance between the two boxes, i.e., IoU loss, cannot optimize the case when the two boxes do not intersect. Second, when two predicted boxes have the same size and the two IoUs are also the same, IoU loss cannot distinguish between the two intersecting cases. To address these problems, GIoU (generalized IoU) loss was proposed in [45], which again improved the model detection performance. However, although IoU loss and GIoU loss are favorable for the IoU metric, there are still problems of slow convergence and inaccurate regression [45]. Additionally, the GIoU loss degenerates into IoU loss when the predicted box is inside the object box and the predicted box is the same size. Thus, this method cannot determine the relative position relationship. To solve these problems, DIoU loss (Distance-IoU) and CIoU loss (Complete IoU) were proposed in [46]. DIoU loss takes into account the overlap area and central point distance between the predicted box and the object box and directly minimizes the normalized distance between the predicted box and the object box. As a result, this method converges much faster than IoU and GIoU loss in training. Based on DIoU loss, CIoU loss takes into account the scale information of the aspect ratio of the bounding box. The CIoU loss integrates the overlap area, central point distance, and aspect ratio of the predicted box and object box, which speeds up the convergence and improves performance.

The mathematical definitions of bounding loss functions are shown below. Definitions of the symbols are in Table 2.

(1)*L*1 loss:
$$\begin{array}{l} L1=|x|\\ \frac{dL1(x)}{x}={\{}_{-1,\;\;\;x\;<\;0}^{1,\;\;\;\;\;\;x\;\geq \;0} \end{array}$$

*x*: The difference between the predicted value and the true value. *L*1 loss is not smooth at the zero point.

(2)*L*2 loss:
$$\begin{array}{l} L2=x^{2}\\ \frac{dL2(x)}{x}=2x \end{array}$$

*L*2 loss has a large *x* value and correspondingly large derivatives at the beginning of training, which makes the initial training unstable.

(3)*SmoothL*1 loss:
$$\begin{array}{l} SmoothL1(x)={\{}_{|x|\;-\;0.5,\;\;\;others}^{0.5x^{2},\;\;\;\;\;\;|x|\;<\;1}\\ \frac{dSmoothL1(x)}{x}={\{}_{\pm 1,\;\;\;others\;}^{x,\;\;\;\;\;\;|x|\;<\;1} \end{array}$$

*SmoothL*1 loss combines *L*1 loss and *L*2 loss, which uses *L*1 loss when *x* is large at the initial stage of training.

(4)*IoU* loss:

*A*: Prediction box; *B*: Ground truth
$$\begin{array}{l} IoU=\frac{A\cap B}{A\cup B}\\ IoU\;loss=1-IoU \end{array}$$

*IoU* loss function for bounding box prediction, which regresses the four bounds of a predicted box as a whole unit.

(5)*GIoU* loss:
$$\begin{array}{l} GIoU=IoU-\frac{\left|C\;-\;A\cup B\right|}{\left|C\right|}\\ GIoU\;loss=1-GIoU \end{array}$$

*C*: The smallest enclosing convex object for *A* and *B*.

When two boxes intersect, *GIoU* takes into account not only the overlapping part but also other non-overlapping parts, which better reflects the overlap of the two boxes.

(6)*DIoU* loss:
$$\begin{array}{l} R_{DIoU}=\frac{\rho^{2}(d_{pred},d_{gt})}{c^{2}}\\ DIoU=IoU-R_{DIoU}\\ DIoU\;loss=1-DIoU \end{array}$$

*DIoU* takes into account not only the overlap area between the prediction box and the ground truth, but also the central point distance.

(7)*CIoU* loss:
$$\begin{array}{l} R_{CIoU}=\frac{\rho^{2}(d_{pred},d_{gt})}{c^{2}}+\alpha \nu \\ v=\frac{4}{\pi^{2}}{\left(\arctan\frac{w_{gt}}{h_{gt}}-\arctan\frac{w}{h}\right)}^{2}\\ \alpha =\frac{v}{(1\;-\;IoU)\;+\;v}\\ CIoU=IoU-R_{CIoU}\\ CIoU\;loss=1-CIoU \end{array}$$

*CIoU* loss takes into account the three geometric factors in bounding box regression: overlap area, central point distance, and aspect ratio.

In this paper, the effects of different bounding box loss functions on model performance are considered based on transfer learning, and the most suitable bounding box loss function, *SmoothL*1, is selected by comparing the experimental results. See Section 3.5 for details.

**Table 2 diagnostics-12-02477-t002:** Definitions of the symbols used in the bounding box function.

Symbol	Explanation
x	The difference between the predicted value and the true value
A	Prediction box
B	Ground truth
C	For A and B, find the smallest enclosing convex object C
$d_{pred}$	Central point of A
$d_{gt}$	Center point of B
ρ	Euclidean distance
c	the diagonal length of the smallest enclosing box covering A and B
α	positive trade-off parameter
ν	Measure the consistency of aspect ratio
$w_{gt}$	The width of B
$h_{gt}$	The height of B
w	The width of A
h	The height of A

### 2.5. The Dataset

In research on cervical cells using deep learning methods, classification and segmentation studies are the most common. Thus, available data sets used are also oriented to these tasks (e.g., Herlev [26], Sipakmed [61], ISBI [62], and AgNOR [63]). In contrast, relatively few studies have performed detection tasks directly on cervical cells, and the datasets are largely private [58]. The cervical cytology image data used in this study were obtained from [35] publicly available datasets (https://github.com/kuku-sichuan/ComparisonDetector (accessed on 12 January 2021)). The dataset includes a total of 7410 cervical microscopical images. There are 6666 images in the training set and 744 images in the test set. The annotation work on the cell images was performed by experienced pathologists. There are 11 categories: ascus (atypical squamous cells of undetermined significance), asch (atypical squamous cells predisposed to high-grade squamous intraepithelial lesions), lsil (low-grade squamous intraepithelial lesion), hsil (high-grade squamous intraepithelial lesion), scc (squamous-cell carcinoma), agc (atypical glandular cells), trich (trichomonas), cand (candida), flora, herps, and actin (actinomyces). Figure 3 shows the number of instances of each category in the training and test sets and the number of images occupied by each category. Figure 4 shows sample images with category annotations.

### 2.6. Experimental Setups

The experiments were conducted based on MM [64]. A warmup strategy was used at the beginning of the model training, and 500 steps were learned with a smaller learning rate. The learning rate gradually increased linearly during the initial 500 iterations, after which the learning rate changed to a pre-set learning rate of 0.0025. Using warmup to increase the learning rate during the initial training steps of the new task, allowing the model to be stabilized under the smaller learning rate. Next, we selected the pre-set learning rate for training after the model was relatively stable, which allowed the model to converge faster and function better. Here, the training epoch is 2× (1× is equivalent to 12 epochs), with 3× epochs of training used if necessary. The learning rate is set to 0.0025, and reduced by a factor of ten after the 8th and 11th epoch, in that order. The batch size is 2, and weight decay and momentum are 0.0001 and 0.9, respectively.

## 3. Results

All experiments were trained on 6666 training images, and the detection performance was based on 744 test images.

### 3.1. Comparison of Transfer Learning for Different Source Data Domains

To determine which is better to use for network model initialization, COCO [43] complete model weights or the ImageNet [42] pre-trained model, relevant experiments were carried out. The experimental data are presented in Table 3. The data show that regardless of whether the backbone is ResNet50 or ResNet101, the network model can obtain higher model detection precision by using the COCO pre-trained model for initialization training. However, when the backbone is ResNet101, the network model uses the COCO pre-trained model to perform initialization training, the final detection result AR decreased slightly. From the experimental result, we can find that AR is more than 20% higher than mAP overall; therefore, here we focus on improving mAP. We choose a higher mAP when the AR difference is not large. It is better to use COCO complete model weights for initialization model detection.

### 3.2. Multi-Scale Training

For experiments without multi-scale training, the default scale used is (1333,800). For multiscale training based on transfer learning, the multi-scale set is [(1333,640), (1333,800)]. The experimental results are shown in Table 4. As shown in Table 4, when the backbone is ResNet50, the best value for detection precision mAP is 60.9% after initialization with COCO’s multi-scale pre-trained model and performing multi-scale training. Compared to the best mAP without multi-scale training (58.7%), the model detection precision mAP improved by 2.2%. The best mAP improved by 2.8 (62.2–59.4)% when the backbone was ResNet101. Therefore, using multi-scale training can improve model detection performance.

### 3.3. Fine-Tuning

The previous experiments all used Frozen_stages = Stem + 1st by default, i.e., the convolutional layer of Stem + 1st stages in the backbone network are frozen during training. To find the setting that can achieve the optimal performance of the model, we conducted fine-tuning experiments, freezing the convolutional layers at different stages of backbone during training, respectively. The results are shown in Table 5. Analyzing the data in the table, we can find that the model performance is optimal when Frozen_stages =Stem + 1st. Meanwhile, we conducted further fine-tuning experiments on network models of different depths (ResNet101), and the results are shown in Table 6. The experimental results show that, again, the model performance is optimal when Frozen_stzges = Stem + 1st. The order of model detection performance in Table 6 is Frozen_stages = (Stem + 1st) > (Stem + first 2) > Stem > no > (Stem + first 3) > (Stem + first 4), which is consistent with the order presented in Table 5. It can be seen that the model detection performance law in the fine-tuning experiments is consistent between network models of different depths. Because the experimental performance when Frozen_stages = Stem + X (X = 2nd, 3rd, 4th) shown in Table 5 did not exceed the performance when Frozen_stages = Stem + 1st, the experiment using Frozen_stages = Stem + X (X = 2nd, 3rd, 4th) in ResNet101 (Table 6) was not conducted anymore.

### 3.4. Only Initialize the Backbone Network Parameters

The previous experiments indicate that network model initialization training using COCO complete model weights yields better model detection performance than the alternative. However, if other researchers were to modify the detection algorithm part of the network model, the COCO complete detection model would not be useable. Considering this factor, we extracted the backbone weight parameters of the COCO complete model and used them to initialize the network model. To determine which works better, using this parameter or using the ImageNet pre-trained model, corresponding experiments were conducted. The experimental results are shown in Table 7. Analyzing the data in the table indicates that the pre-trained network model offered better model detection performance than those without pre-training, and initial training with the backbone weight parameters of the complete COCO model offered better model detection performance than initial training with ImageNet pre-training. This result also provides a reference for other researchers. When some modifications to the detection algorithm in the network model make it impossible to use the COCO complete model weights, then it is possible to use only the backbone weight parameters of the COCO complete model to perform initialization training in the network model for the current task.

### 3.5. Different Bounding Box Loss Experiments

Relevant experiments were performed on existing COCO pre-trained models with different bounding box losses. The experimental data are shown in Table 8. The experimental results indicate that the model detection precision is not improved by directly replacing the loss function with IoU and GIoU loss. The highest model detection precision is achieved when the loss function is SmoothL1. The pre-trained model “Faster_rcnn_r101 _fpn_mstrain_3x_coco.pth” is obtained when the bounding box loss is L1 loss, which is used to optimize the four points of the prediction box. SmoothL1 loss combines the advantages of L1 loss and L2 loss, and also optimizes the object box as four independent variables, whereas IoU and GIoU loss considers the prediction box as a whole. This method uses the same bounding box loss function as the corresponding type of pre-trained model, which helps improve the performance of model detection. Next, we conducted validation experiments. Since IoU loss and GIoU loss had no corresponding COCO 3x (36epochs) multi-scale pre-training model, for the sake of fairness, pre-training models used the COCO 1x (12epochs) pre-training model corresponding to the loss function. The experimental data are shown in Table 9. The data in the table indicate that the model detection precision is highest when the bounding box loss is GIoU. Here, the detection precision of the model with SmoothL1 loss, IoU, and GIoU loss exceeds that of the model with L1 loss. Thus, the network model uses a bounding box loss function of the same type as that of the pre-trained model, which facilitates optimization of the network model and improves model detection performance. IoU and GIoU do not have corresponding COCO multi-scale training models. The data in Table 8 show that when multi-scale training is performed, the model detection precision is highest when SmoothL1 is used for the bounding box loss function. Therefore, we utilize the SmoothL1 bounding box loss function.

### 3.6. Using Different Means and Stds for Multi-Scale Training

So far, all experiments have used ImageNet’s mean and std by default. To determine which mean and std will result in the best model performance when normalizing the input images, one method is to use the mean and std of the corresponding data set of the pre-training, and the other method is to use the mean and std of the data set used in the experiment. To determine which method is better, corresponding experiments were carried out. The experimental data are shown in Table 10. By analyzing the results, we found that the model performance performed best when using the mean and std from the cervical cell data set used in the experiment.

When the backbone is ResNet50, the optimal mAP is 61.6% and the detection result images are shown in Figure 5.

### 3.7. The Results with State-of-the-Art Methods

Based on the state-of-the-art detection algorithm Faster R-CNN [48], we explored the effects of the transfer learning of different source data domains, multi-scale training, the fine-tuning strategy, the bounding box loss function, and different means and stds on the detection performance of the model in cervical cells/clumps. Finally, the detection performance of the model was significantly improved. To further validate the effectiveness of the method used in our experiments, we applied the method to another state-of-the-art algorithm, RetinaNet [56]. The results of the experiments are shown in Table 11. Other experimental results are also listed in the table for the current data set. As shown in Table 11, when using on both Faster R-CNN and RetinaNet, our method improved the detection performance value of both mAP and AR.

## 4. Discussion

Based on a survey study of cervical cancer detection using deep learning, this study explored using the COCO pre-trained model for initialization training of the network model instead of the more commonly used ImageNet pre-trained model.

The important findings in our study are as follows. (1) With the current cervical cell dataset, the network model offered better detection performance using COCO complete model weights for initialization compared to using ImageNet for initialization. (2) The network model adopted a multi-scale training approach, which was able to improve the robustness of the model to cells at different scales and contribute to the improvement of the model detection performance. (3) When the backbone weight parameters of the COCO complete model and ImageNet pre-trained model were used, respectively, to initialize the parameters of the network model backbone part, the former was found to offer better detection performance, which will also provide a reference for other researchers. (4) Considering the impact of different bounding box losses on network model performance and analyzing the impacts of different bounding box losses on the experimental results, we found that using a bounding box loss function consistent with the type of pre-trained model contributed to the model detection performance. (5) Normalization of the input images using the mean and std of the cervical cell dataset used in the current experiment resulted in better model detection performance.

In our study, the fine-tuning method was used to find the best performance of the model. At the same time, to demonstrate that some conclusions in the experiment apply to network models of different depths, we conducted experiments on network models based on backbone ResNet50 and ResNet101. We also validated the effectiveness of our method on another state-of-the-art detection algorithm, RetinaNet. The final model detection precision based on backbone Resnet50 was 61.6% for mAP and 87.7% for AR, indicating a 12.8% improvement in mAP and 23.7% improvement in AR compared to the previous results of the dataset used in this study, and the model performance was greatly improved. This study will have notable reference significance and value for other researchers.

## Figures and Tables

**Figure 1 diagnostics-12-02477-f001:**
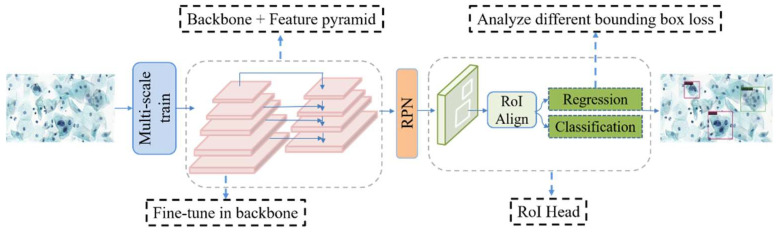
Overview of the network model. In the figure, the RPN (Region Proposal Network) and ROI Head (Region of Interest) represent the first and second stages of Faster R-CNN.

**Figure 2 diagnostics-12-02477-f002:**
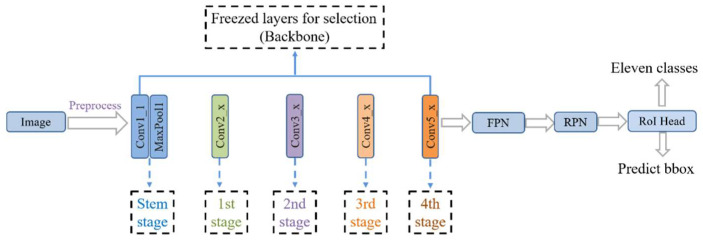
Network freezing model.

**Figure 3 diagnostics-12-02477-f003:**
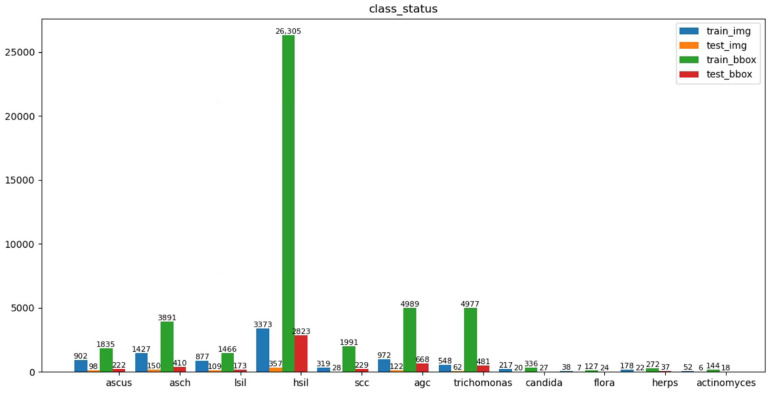
Distribution of categorical instances on the training and test sets.

**Figure 4 diagnostics-12-02477-f004:**
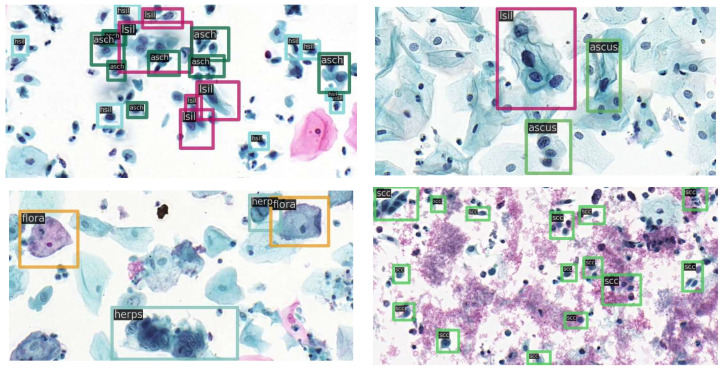
Sample images with category annotations. Different colors of the bounding box represent different categories.

**Figure 5 diagnostics-12-02477-f005:**
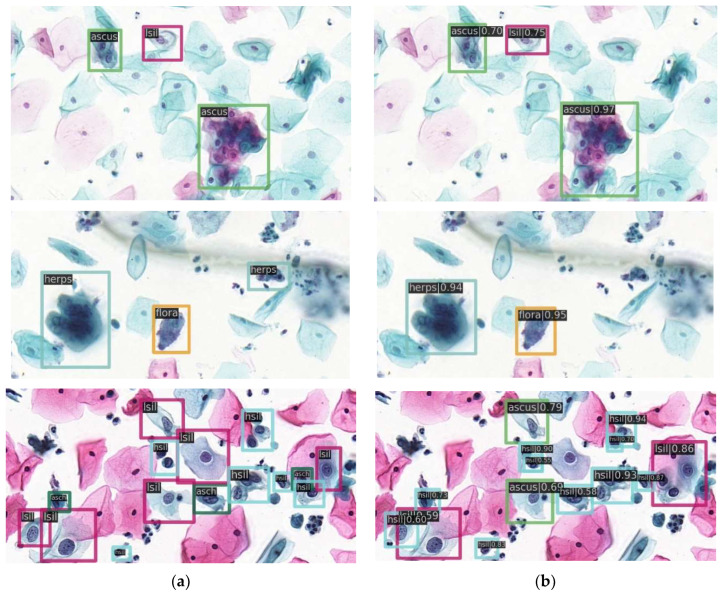
Detection results on the test image. (**a**) Left images are test images with ground truth; (**b**) right images are the predicted images. Different colors of the bounding box represent different categories.

**Table 1 diagnostics-12-02477-t001:** The procedure of resizing the input image.

Resize Input Image
(1) Calculate the large and small values of the random selection scale. **max-edge = max(scale)** **min-edge = min(scale)**
(2) Calculate the resizing ratio (scale-factor): (max-edge)/long side, (min-edge)/short side, and choose a small ratio (make the image as close to the original size as possible while being resized). **scale-factor = min(max-edge/max(h, w), min-edge/min(h, w))**
(3) Original image width: w, height: h. Calculate the width and height of the image after resizing. **scale-w = int(w × float(scale-factor) + 0.5)** **scale-h = int(h × float(scale-factor) + 0.5)**
(4) After resizing, the padding operation is performed, and the resized image is padded to a multiple of size-divisor = 32 to avoid feature loss during convolution. The width and height of the final image are pad-w, and pad-h. **pad-w = int(np.ceil (scale-w/size-divisor)) × size-divisor** **pad-h = int(np.ceil (scale-h/size-divisor)) × size-divisor**

**Table 3 diagnostics-12-02477-t003:** Experimental results of using COCO and ImageNet pre-training. Here, “Faster_rcnn_r50_ fpn_1x.pth” represents the COCO pre-trained model weights; Faster_ rcnn is the detection algorithm; r50 represents ResNet50; and x represents 1 × 12 epochs (1x = 12 epochs).

Backbone	Initialization	mAP (%)	AR (%)
ResNet50	None	8.2	41.9
ResNet50	ImageNet	51.9	83.3
ResNet50	Faster_rcnn_r50_fpn_1x_coco.pth	58.4	84.1
ResNet50	Faster_rcnn_r50_fpn_2x_coco.pth	57.6	85.9
ResNet101	None	7.2	37.5
ResNet101	ImageNet	58.3	86.0
ResNet101	Faster_rcnn_r101_fpn_1x_coco.pth	58.5	84.7
ResNet101	Faster_rcnn_r101_fpn_2x_coco.pth	59.8	84.0

**Table 4 diagnostics-12-02477-t004:** The detection results of the model for multi-scale training. “Faster_rcnn_r50 _fpn_mstrain _3x_coco.pth” is the multi-scale pre-trained model of COCO; mstrain: multi-scale training.

Backbone	mstrain	Initialization	mAP (%)	AR (%)
ResNet50	n	Faster_rcnn_r50_fpn_mstrain_3x_coco.pth	58.7	85.7
ResNet50	y	Faster_rcnn_r50_fpn_mstrain_3x_coco.pth	60.9	87.2
ResNet50	y	Faster_rcnn_r50_fpn_1x_coco.pth	60.0	86.3
ResNet50	y	Faster_rcnn_r50_fpn_2x_coco.pth	60.5	86.4
ResNet101	n	Faster_rcnn_r101_fpn_mstrain_3x_coco.pth	59.4	85.2
ResNet101	y	Faster_rcnn_r101_fpn_mstrain_3x_coco.pth	62.2	86.9
ResNet101	y	Faster_rcnn_r101_fpn_1x_coco.pth	61.3	88.2
ResNet101	y	Faster_rcnn_r101_fpn_2x_coco.pth	61.7	87.3

**Table 5 diagnostics-12-02477-t005:** Fine-tuning experiments of the model with ResNet50 backbone were performed based on transfer learning and multi-scale training.

Backbone	Frozen_Stages	mAP (%)	AR (%)	Params (M)
ResNet50	no	59.4	87.1	41.4
ResNet50	Stem	59.5	87.4	41.39
ResNet50	Stem + 1st	60.9	87.2	41.17
ResNet50	Stem + first 2	59.9	86.6	39.95
ResNet50	Stem + first 3	54.9	84.9	32.86
ResNet50	Stem + first 4	49.4	84.5	17.89
ResNet50	Stem + 2nd	59.4	87.4	40.17
ResNet50	Stem + 3rd	58.5	86.7	34.29
ResNet50	Stem + 4th	58.8	86.9	26.43

**Table 6 diagnostics-12-02477-t006:** Fine-tuning experiments of the model with the ResNet101 backbone were performed based on transfer learning and multi-scale training.

Backbone	Frozen_Stages	mAP (%)	AR (%)	Params (M)
ResNet101	no	60.2	87.2	60.39
ResNet101	Stem	61.5	86.8	60.38
ResNet101	Stem + 1st	62.2	86.9	60.17
ResNet101	Stem + first 2	61.9	86.4	58.95
ResNet101	Stem + first 3	55.7	85.3	32.86
ResNet101	Stem + first 4	46.5	81.6	17.89

**Table 7 diagnostics-12-02477-t007:** Initialization training of the network model backbone component using different pre-training parameters (when conducting multi-scale training).

Backbone	Initialization	mAP (%)	AR (%)
ResNet50	None	8.9	43.2
ResNet50	ImageNet	57.4	86.0
ResNet50	faster_rcnn_r50_fpn_mstrain_3x_coco_only_backbone.pth	57.5	86.5
ResNet101	None	8.4	42.7
ResNet101	ImageNet	57.9	87.0
ResNet101	faster_rcnn_r101_fpn_mstrain_3x_coco_only_backbone.pth	59.8	86.8

**Table 8 diagnostics-12-02477-t008:** To further improve the detection performance of the model, we set different bounding box losses for the experiments. Bbox loss: Bounding box loss. (Conducting multi-scale Training).

Model	Initialization	Bbox Loss	mAP (%)	AR (%)
ResNet50	Faster_rcnn_r50_fpn_mstrain_3x_coco.pth	L1	60.9	87.2
ResNet50	Faster_rcnn_r50_fpn_mstrain_3x_coco.pth	SmoothL1	61.1	86.9
ResNet50	Faster_rcnn_r50_fpn_mstrain_3x_coco.pth	IoU	60.4	87.3
ResNet50	Faster_rcnn_r50_fpn_mstrain_3x_coco.pth	GIou	60.1	87.9

**Table 9 diagnostics-12-02477-t009:** Relevant experiments were conducted to verify whether using bounding box loss consistent with the type of pre-trained model contributed to an improvement of model detection performance.

Model	Initialization	Bbox Loss	mAP(%)	AR(%)
ResNet50	Faster_rcnn_r50_fpn_1x _coco.pth	L1	58.4	84.1
ResNet50	Faster_rcnn_r50_fpn_1x _coco.pth	SmoothL1	59.2	84.3
ResNet50	Faster_rcnn_r50_fpn_iou_1x_coco.pth	IoU	59.0	86.1
ResNet50	Faster_rcnn_r50_fpn_giou_1x_coco.pth	GIou	59.7	84.8

**Table 10 diagnostics-12-02477-t010:** Multi-scale training with the means and stds of different data. Initialization: using Faster_ rcnn_r50_fpn_mstrain_3x_coco.pth; Box loss: Bounding box loss; self: denotes the cervical cell data set used for the experiments.

Backbone	Box Loss	Mean, std from Data	mAP (%)	AR (%)
ResNet50	SmoothL1	ImageNet	61.1	86.9
ResNet50	SmoothL1	coco	61.2	87.7
ResNet50	SmoothL1	self	61.6	87.7

**Table 11 diagnostics-12-02477-t011:** The performance of different methods. The backbone is ResNet50. Initialization: The initialization type used by the model.

Model	Comparison Detector [35]	Faster R-CNN [46] [45]	* Faster R-CNN	RetinaNet [56]	* RetinaNet
Initialization	ImageNet	ImageNet	COCO	ImageNet	COCO
mAP(%)	48.8	51.9	61.6	53.8	57.2
AR(%)	64	83.3	87.7	81.4	88.3

“*” indicates the test result after using our method.

## Data Availability

The image data used in the study are openly accessible at the following site: https://github.com/kuku-sichuan/ComparisonDetector (accessed on 12 January 2021).

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
