# Peer review of "Cervical Cell/Clumps Detection in Cytology Images Using Transfer Learning"

_diagnostics, 2022, doi:10.3390/diagnostics12102477_

Round 1
Reviewer 1 Report
In the study, the authors used COCO pretrained models for cervical cell detection to overcome the limited data set issue at training time. For further optimization the investigators tested also different strategies of fine-tuning, multi-scale training, and metrics for bounding-box loss. It appears that the proposed transfer learning approach can significantly improve the cell detection accuracy. The manuscript can be improved by addressing the following issues:
1) The overall writing is readable. However, it is quite cumbersome in many different places and can be much more concise.
2) A major limitation of the study is that the performance assessment was solely based on a single dataset. Experimental results from multiple datasets would be more convincing. The authors should consider testing their models with other open-source datasets. If a model performance is only optimized for a fixed dataset, its clinical application would be quite limited.
3) The descriptions for transfer learning on P4 are quite general. The authors should provide more specifics about the precise procedures used for the fine-tuning, such as the frozen CNN layers in the backbone. A part of the information is mislocated in the result section. For clarity, the different frozen stages (0-4) should be explicitly annotated in Figure 2.
4) It seems that the initialization of the backbone with the weights from training using the COCO or ImageNet datasets makes a big difference in the model performance. I recommend moving the section 3.4 to 3.1. That is to present the most important result first. Other improvements are quit minor.
5) As shown in Tables 4 and 5, the mAP was reduced sharply with frozen_stages 3 and 4, indicating the importance to fine-tune parameters in the conv4 and 5 layers. On the other hand, the mAP results for the cases of “Only 3” and 4 (Table 4) are quite decent. Could the authors provide a reasonable explanation for this?
Author Response
Response to Reviewer 1 Comments
Point 1: The overall writing is readable. However, it is quite cumbersome in many different places and can be much more concise.
Response 1: Thank you very much for your suggestion, we have re-examined the paper carefully and revised the cumbersome part to make it as concise as possible.
Point 2: A major limitation of the study is that the performance assessment was solely based on a single dataset. Experimental results from multiple datasets would be more convincing. The authors should consider testing their models with other open-source datasets. If a model performance is only optimized for a fixed dataset, its clinical application would be quite limited.
Response 2:
During the study, we found that the datasets used for cervical cell detection were largely private (we also mentioned this in the introduction section of the dataset in the paper), we tried our best to find publicly available datasets for cervical cell detection, and only one [1] was obtained so far.
We did the research again, and the latest survey [2] summarizes the publicly available databases in cervical cytology images, we can also find that most publicly available datasets are used for cervical cell classification and segmentation tasks, and the only publicly available one performing detection tasks on cervical cell images is CDetector [1], the survey table we have placed at the appendix below
We will conduct further research if a suitable dataset becomes available in the future. We are also building our dataset now.
Point 3: The descriptions for transfer learning on P4 are quite general. The authors should provide more specifics about the precise procedures used for the fine-tuning, such as the frozen CNN layers in the backbone. A part of the information is mislocated in the result section. For clarity, the different frozen stages (0-4) should be explicitly annotated in Figure 2.
Response 3: Thank you very much for your suggestion, we have made changes to the transfer learning section. We have made a specific description of the Fine-tuning procedure. We have redrawn Figure 2, the different frozen stages (0-4) have been explicitly annotated in Figure 2.
Point 4: It seems that the initialization of the backbone with the weights from training using the COCO or ImageNet datasets makes a big difference in the model performance. I recommend moving the section 3.4 to 3.1. That is to present the most important result first. Other improvements are quit minor.
Response 4: We appreciate your suggestion, and we have considered it carefully. However, our logical sequence is this: section 3.1 has not yet performed multi-scale training, section 3.2 proves that multi-scale training can improve model detection performance, and section 3.4 performs multi-scale training. Therefore, from this viewpoint, we still follow the original logical structure. We still appreciate your suggestion.
Point 5: As shown in Tables 4 and 5, the mAP was reduced sharply with frozen_stages 3 and 4, indicating the importance to fine-tune parameters in the conv4 and 5 layers. On the other hand, the mAP results for the cases of “Only 3” and 4 (Table 4) are quite decent. Could the authors provide a reasonable explanation for this?
Response 5:
We have redrawn Figure 2 for a clearer representation of the different frozen stages. Frozen _stages=3 in the original Table 4 means frozen stem+1st+2nd+3rd stages; frozen_stages=4 means frozen stem+1st+2nd+3rd+4th stages (the entire backbone network). The "Only 3" and 4 in the original Table 4 indicate frozen stems+3rd and stem+4th stages, respectively.
When freezing the stem+first 3 or stem+first 4 stages, the number of parameters decreases more, and the network learning ability is reduced sharply, the mAP is reduced sharply.
When freezing stem+ 3rd and stem+ 4th, although the number of parameters also decreases a considerable amount, the other layers in the backbone are still in action and the network model still has a large learning capacity, so the mAP results are quite decent. For example, when the stem+4th stages are frozen, the 1st and 2nd and 3rd stages in the backbone network still act in the network.
Figure 2. Network freezing model
Reference
- Liang, Yixiong, et al. "Comparison detector for cervical cell/clumps detection in the limited data scenario." Neurocompu 499 ting 437 (2021): 195-205.
- Jiang, Hao, et al. "Deep Learning for Computational Cytology: A Survey." arXiv preprint arXiv:2202.05126 (2022).
Appendix:

Reviewer 2 Report
Major revision
The paper is well-written and well-organised. However there are a few points which the authors have not addressed.
1. The motivation towards selecting the architecture is not clear
2. The paper lacks explainablity and mathematical validation towards the selection of the model and loss functions.
3. Comparision with state of the art segmentation models such as U-Net, LinkNet in place of FPN have not been discussed.
Suggestions:
3. The authors are suggested to add better representation of the outputs provided by the proposed model.
4. The authors are suggested to add an ablation study wherein the change in accuracy scores (mAP and AR) with respect to the number of data-samples are mentioned.
5. The literature review is very trivial. The authors are encouraged to add the paper:https://pubmed.ncbi.nlm.nih.gov/34328530/
https://www.sciencedirect.com/science/article/pii/S0010482522007910?dgcid=rss_sd_all
Author Response
Response to Reviewer 2 Comments
Point 1: The motivation towards selecting the architecture is not clear.
Response 1:
Due to the amount of data, we chose to use transfer learning, and again considering the differences between detection and classification tasks, we proposed to use the COCO pre-trained model. We mentioned this in the abstract section of the paper, and I apologize for not making you understand that, this is my problem, and I have made some changes in the abstract section of the paper.
Our experiments also demonstrate the effectiveness of using the COCO pre-trained model to do transfer learning in this study.
Point 2: The paper lacks explainablity and mathematical validation towards the selection of the model and loss functions.
Response 2: Thank you very much for your suggestion. We have supplemented the content in our paper.
Point 3: Comparision with state of the art segmentation models such as U-Net, LinkNet in place of FPN have not been discussed.
Response 1: Since the dataset used in our study is dedicated to the detection task, it is not able to perform the segmentation task.
Suggestion:
Point 3: The authors are suggested to add better representation of the outputs provided by the proposed model.
Response 3: Thank you very much for your suggestion, we have changed the output representation of the model to be better and clearer.
Point 4: The authors are suggested to add an ablation study wherein the change in accuracy scores (mAP and AR) with respect to the number of data-samples are mentioned.
Response 4: We have carefully considered this proposal from the following aspects.
(1) The dataset used in this study has a small sample size for some categories, so it is not feasible to reduce some samples, which could easily result in missing categories.
(2) Some cervical cell images have multiple categories of ground truth, and it is difficult to reduce the sample size in the same proportion.
(3) Importantly, the purpose of our experiments is to demonstrate that transfer learning works well in the current dataset. In addition, from a deep learning perspective, as the amount of data decreases, the performance will decrease, so it is less meaningful to do experiments.
(4) For our approach, the sample size is not the focus of the study.
(5) Thank you very much for your suggestion, and we will discuss this issue in depth again if a suitable dataset becomes available in the future.
Point 5: The literature review is very trivial. The authors are encouraged to add the
paper:https://pubmed.ncbi.nlm.nih.gov/34328530/
https://www.sciencedirect.com/science/article/pii/S00104825220
07910?dgcid=rss_sd_all
Response 5:
We thank you for your suggestions and we have read the two papers carefully.
We cited the paper: https://www.sciencedirect.com/science/article/pii/S0010482522007910?dgcid =rss_sd_all in the Introduction of our paper and added it to the reference list. By reading the paper: https://pubmed.ncbi.nlm.nih.gov/34328530/", we found that the paper uses machine learning (ML) algorithms. Our study is based on deep learning algorithms, so the paper is not cited.
We thank you again for your suggestions, which have been very helpful to our study.

Round 2
Reviewer 1 Report
New commnets are highlighted in purple.

Author Response
Response to Reviewer 1 Comments
Point 1: The overall writing is readable. However, it is quite cumbersome in many different places and can be much more concise.
Response 1: Thank you very much for your suggestion, we have re-examined the paper carefully and revised the cumbersome part to make it as concise as possible.
Comments: The revised manuscript with annotations is difficult to read. I cannot assess the quality
of the rewritting.
Response to Comments: We apologize for making your reading difficult. It is our problem that the changes were not clearly indicated, and we have listed our changes below.
(1) Abstract section: in line 17-19(page 1) of revised manuscript, we have changed the original “such studies used the same transfer learning method initialized by the ImageNet pre-trained model on the backbone network in two different types of tasks for the detection and classification of cervical cells” to the current “such studies used the same transfer learning method that is the backbone network initialization by the ImageNet pre-trained model in two different types of tasks, the detection and classification of cervical cell/clumps”
(2) Abstract section: in line 30-31(page1) of revised manuscript, we have changed the original “when using a network model based on the ResNet50 Backbone, the mAP (mean Average Precision) was found to be 61.6% and the AR (Average Recall) was 87.7%” to the current “based on backbone Resnet50, the mean Average Precision (mAP) of the network model is 61.6% and Average Recall( AR) is 87.7%.”
(3) Intruction section: in line 81-83(page2) of revised manuscript, we have changed the original “Extensive experiments were conducted to demonstrate that network models in cervical cell detection studies perform better when using COCO [40] pre-trained model parameters for initialization compared to using ImageNet [39] pre-trained model parameters for initialization.” to the current “This study conducted extensive experiments to demonstrate that network model initialization using COCO [43] pre-trained models was better than using ImageNet [42] pre-trained models in the cervical cell detection study.”
(4) Intruction section: in line 87-90(page2) of revised manuscript, we have changed the original “In this study, to improve the robustness of the network model in detecting cells of different scales and to improve the detection precision of the model, we perform multi-scale training according to the actual situation of the dataset and transfer learning.” to the current “To improve the robustness of the network model to cervical cells of different scales, In this study, multi-scale training is carried out according to the actual situation of the dataset and transfer learning.”
(5) Intruction section: in line 91-93(page2) of revised manuscript, we have changed the original “Bounding box loss is very important for the accurate localization of object detection, and the bounding box loss function has been optimized in recent years (see the optimization history for bounding box loss: L1, L2 loss; SmoothL1 loss; IoU loss [41]; GIoU loss [42]; DIoU loss [43]; CIoU loss [43] ).” to the current “Bounding box loss is very important for the accurate localization of object detection, and the optimization history of bounding box loss in recent years: L1, L2 loss; SmoothL1 loss; IoU loss [44]; GIoU loss [45]; DIoU loss [46]; CIoU loss [46].”
(6) 2.2.transfer learning section: in line 156(page4) of revised manuscript, we have changed the original “The dataset used in this study includes 7410 images (6666 images in the training set and 744 images in the test set),” to the current “The dataset used in this study includes 6666 images in the training set,”
(7) 2.2.transfer learning section: in line 158-159(page4) of revised manuscript, we have changed the original “Therefore, according to the actual situation, we selected the third way to fine-tune the network model and analyze the impact of using the COCO pre-trained model and ImageNet pre-trained model to perform transfer learning on the model performance.” to the current “Therefore, according to the actual situation, we selected the third way to fine-tune the network model.”
(8) 2.2.transfer learning section: in line 161-162(page4) of revised manuscript, we have changed the original “Figure 2 shows a graph of the parameter freezing model used in this study.” to the current “Figure 2 shows the network freezing model of the study.”
(9) 3.1.Comparison of transfer learning for different source data domains section: in line 312-314(page9) of revised manuscript, we have changed the original “Ultimately, the final detection result AR decreased slightly. The experimental result AR was 20% higher than mAP overall, so here we focused on improving mAP.” to the current “the final detection result AR decreased slightly. From the experimental result, we can find that AR is more than 20% higher than mAP overall, so here we focus on improving mAP”。
(10) 3.3.Fine-tuning section: in line 333-334(page10) of revised manuscript, we have changed the original “the stem of ResNet (see Figure 2). The first stages were trained with frozen parameters.” to the current “the convolutional layer of Stem+1st stages in the backbone network are frozen during training.”。
(11) 3.3.Fine-tuning section: in line 335-337(page10) of revised manuscript, we have changed the original “To find the settings able to obtain optimal performance in the model, we conducted fine-tuning experiments, in which different stages of ResNet were frozen separately during the training phase.” to the current “To find the setting that can achieve the optimal performance of the model, we conducted fine-tuning experiments, freezing the convolutional layers at different stages of backbone during training, respectively.”
(12) 3.3.Fine-tuning section: in line 337-338(page11) of revised manuscript, we have changed the original “Analyzing the data in the table shows that the model performance is optimal when Frozen_stages=1.” to the current “Analyzing the data in the table, we can find that the model performance is optimal when Frozen_stages =Stem+1st.”
(13) In Table 5 and Table 6(page11), we make the representation of "Frozen_stages" in the table more clear。
(14) 3.6.Using different means and stds for multi-scale training section: in line 401 of revised manuscript, we have changed the original “The experiments conducted so far used the default ImageNet mean and std.” to the current “So far, all experiments have used ImageNet's mean and std by default.”
(15) 4.Discussion section: in line 450-451(page15) of revised manuscript, we have changed the original “the fine-tuning method was used to find the best performance of the model according to the actual situation of the dataset.” to the current “In our study, the fine-tuning method was used to find the best performance of the model.”
Point 2: A major limitation of the study is that the performance assessment was solely based on a single dataset. Experimental results from multiple datasets would be more convincing. The authors should consider testing their models with other open-source datasets. If a model performance is only optimized for a fixed dataset, its clinical application would be quite limited.
Response 2:
During the study, we found that the datasets used for cervical cell detection were largely private (we also mentioned this in the introduction section of the dataset in the paper), we tried our best to find publicly available datasets for cervical cell detection, and only one [1] was obtained so far.
We did the research again, and the latest survey [2] summarizes the publicly available databases in cervical cytology images, we can also find that most publicly available datasets are used for cervical cell classification and segmentation tasks, and the only publicly available one performing detection tasks on cervical cell images is CDetector [1], the survey table we have placed at the appendix below
We will conduct further research if a suitable dataset becomes available in the future. We are also building our dataset now.
Comments: Incorrect. There are certainly more openly accessible datasets available for cervical cell
detection. For example, dataset used in ref 38 ( X. Li et al) is definitely openly accessible. As pointed
out in the cent nnU-net study (https://www.nature.com/articles/s41592-020-01008-z), It is an important criterion to test a deep-learning model with multiple datasets. Since a deep learning model
invovle typically millions of parameters and a single datset may only optimize a small fraction of the
parameters.
Response to Comments:
Thank you very much for the reminder. We previously found in the “Data Availability Statemen” of ref38 (X. Li et al) that the authors provided a link to access the dataset at “https://tianchi.aliyun.com /competition/entrance/231757/ introduction. “. We clicked to find out that this was a competition dataset and that the dataset file was only available for download if you signed up for the competition(Figure 1-1). But unfortunately, the competition is already finished (Figure 1-2).
We now also did our best to find this dataset and found that the dataset is very large, with a single image size of 300~400M and the whole dataset over 200G. Because the single image is too large, it is impractical to take it directly for training, so it also needs to crop the image, and the whole dataset is again very large, and it will take a long time if we go to do it because of our limited equipment and experimental environment.
In addition, as a case study, we use transfer learning in our study to improve the detection performance of the model for the current dataset, considering the volume of the dataset. But the amount of data in this competition dataset is large, and when the amount of data is sufficient, using the dataset itself to train the model may achieve very good results.
As a case study, we obtain the optimal detection precision on the current dataset through fine-tuning and a series of methods. Also this is the limitation of our paper, just as different nails fit different hammers, different data sets have different tuning methods. In future study, we will go to work to find a universally applicable method.
Figure 1-1
Figure 1-2
Point 3: The descriptions for transfer learning on P4 are quite general. The authors should provide more specifics about the precise procedures used for the fine-tuning, such as the frozen CNN layers in the backbone. A part of the information is mislocated in the result section. For clarity, the different frozen stages (0-4) should be explicitly annotated in Figure 2.
Response 3: Thank you very much for your suggestion, we have made changes to the transfer learning section. We have made a specific description of the Fine-tuning procedure. We have redrawn Figure 2, the different frozen stages (0-4) have been explicitly annotated in Figure 2.
Point 4: It seems that the initialization of the backbone with the weights from training using the COCO or ImageNet datasets makes a big difference in the model performance. I recommend moving the section 3.4 to 3.1. That is to present the most important result first. Other improvements are quit minor.
Response 4: We appreciate your suggestion, and we have considered it carefully. However, our logical sequence is this: section 3.1 has not yet performed multi-scale training, section 3.2 proves that multi-scale training can improve model detection performance, and section 3.4 performs multi-scale training. Therefore, from this viewpoint, we still follow the original logical structure. We still appreciate your suggestion.
Comments: The results indicate that multi-scale train contributes about 2% improvement in mAP, transfer learning contributes over 20% improvement in mAP, and bbox-loss optimization contributes Less than 0.5% improvement in mAP. It should be very clear what is most significant.
Response to Comments:
We are very grateful for your suggestion and we understand what you mean. We have again carefully reviewed our paper and thought about this issue carefully. In section 3.1, our aim is to go find the better transfer source. But we really should first prove that transfer learning is valid in the current dataset, which is our problem. Therefore, we added two experiments in Table3: the experiments without transfer learning when bacbone is ResNet50 and ResNet101, at which the mAP is 8.2%, 7.2%, respectively. From Table 3, it is clear that the detection precision of the model is greatly improved when transfer learning is performed. That is, transfer learning is effective.
Now, the whole logical structure is more smooth: in Section 3.1, we have proved that transfer learning is effective and selected the appropriate transfer source; In section 3.2, it is proved that multi-scale training is effective; In section 3.4, Multi-scale training is used . This is logical.
Table 3
|
Backbone |
Initialization |
mAP(%) |
AR(%) |
|
ResNet50 |
None |
8.2 |
41.9 |
|
ResNet50 |
ImageNet |
51.9 |
83.3 |
|
ResNet50 |
Faster_rcnn_r50_fpn_1x_coco.pth |
58.4 |
84.1 |
|
ResNet50 |
Faster_rcnn_r50_fpn_2x_coco.pth |
57.6 |
85.9 |
|
ResNet101 |
None |
7.2 |
37.5 |
|
ResNet101 |
ImageNet |
58.3 |
86.0 |
|
ResNet101 |
Faster_rcnn_r101_fpn_1x_coco.pth |
58.5 |
84.7 |
|
ResNet101 |
Faster_rcnn_r101_fpn_2x_coco.pth |
59.8 |
84.0 |
Point 5: As shown in Tables 4 and 5, the mAP was reduced sharply with frozen_stages 3 and 4, indicating the importance to fine-tune parameters in the conv4 and 5 layers. On the other hand, the mAP results for the cases of “Only 3” and 4 (Table 4) are quite decent. Could the authors provide a reasonable explanation for this?
Response 5:
We have redrawn Figure 2 for a clearer representation of the different frozen stages. Frozen _stages=3 in the original Table 4 means frozen stem+1st+2nd+3rd stages; frozen_stages=4 means frozen stem+1st+2nd+3rd+4th stages (the entire backbone network). The "Only 3" and 4 in the original Table 4 indicate frozen stems+3rd and stem+4th stages, respectively.
When freezing the stem+first 3 or stem+first 4 stages, the number of parameters decreases more, and the network learning ability is reduced sharply, the mAP is reduced sharply.
When freezing stem+ 3rd and stem+ 4th, although the number of parameters also decreases a considerable amount, the other layers in the backbone are still in action and the network model still has a large learning capacity, so the mAP results are quite decent. For example, when the stem+4th stages are frozen, the 1st and 2nd and 3rd stages in the backbone network still act in the network.
Comments: The question was not answered. According to the results shown in Tables 5 and 6 (new),
It does not that much if stem, conv1 and 2 were fined or not and the mAP was barely influenced.
Howeer, if conv4 was not fine tuned, the mAP was reduced by about 5% (ResNet50) to 10%
(ResNet101). If conv3 was not fined-tuded, the mAP was reduced by about 5%. This means that fine-
tunning of conv3 and 4 are very important. On the other hand, the results for Fronzen stem+3 and
Fronzen+4 (see Table 5) are quite decent and the mAP was about 59% (not significantly different from
the hightest mAP), suggesting that the fine-tuning of conv3 and 4 are not so vital. The authors need
to provide a reasonable explanation for the apparely two conflicting observations.
Response to Comments:
In order to make the different freezing levels in the table more clearly represented, we again modified the relevant contents of the Frozen_stages column in Table 5 and Table 6.
Frozen _stages=Stem+first 2 in Table 5 means frozen Stem+1st+2nd stages; Frozen _stages= Stem+first 3 in Table 5 means frozen stem+1st+2nd+3rd stages; Frozen _stages=Stem+first 4 in Table 5 means frozen stem+1st+2nd+3rd+4th stages. The data in Table 5 are shown below, and Table 5 is used as an example for analysis.
Table 5
|
Backbone |
Frozen_stages |
mAP(%) |
AR(%) |
Params(M) |
|
ResNet50 |
no |
59.4 |
87.1 |
41.4 |
|
ResNet50 |
Stem |
59.5 |
87.4 |
41.39 |
|
ResNet50 |
Stem+1st |
60.9 |
87.2 |
41.17 |
|
ResNet50 |
Stem+first 2 |
59.9 |
86.6 |
39.95 |
|
ResNet50 |
Stem+first 3 |
54.9 |
84.9 |
32.86 |
|
ResNet50 |
Stem+first 4 |
49.4 |
84.5 |
17.89 |
|
ResNet50 |
Stem+2nd |
59.4 |
87.4 |
40.17 |
|
ResNet50 |
Stem+3rd |
58.5 |
86.7 |
34.29 |
|
ResNet50 |
Stem+4th |
58.8 |
86.9 |
26.43 |
From Table 5 we can find the mAP was barely influenced when Frozen_stages=Stem, Stem+1st and Stem+first 2. We think Params is a big factor, Because the data in the table shows that the difference between Params is very small at this time. When Frozen_stages=Stem+first 3, Params drop by about 8M (RestNet50) and mAP is reduced by about 5% (ResNet50). When Frozen_stages =Stem+first 4, the Params drop by about 23M (ResNet50) and the mAP is reduced by about 10%.
But at the same time, we can see that when Frozen_stages=Stem+3rd and Frozen_stages= Stem+4th, the params decreases a considerable amount, but the mAP is quite decent. So we think that at this time there are other factors besides the influence of params. For example, when Frozen_stages =Stem+3rd, the params also dropped a considerable amount, but at this time the 1st+2nd+4th in the backbone did not be frozen and the network model still has a large learning capacity, so the mAP are quite decent.
In summary, we believe that if we consider the impact of freezing different layers on model performance, the number of params is a large factor, but there are other complex factors in play. We plan to continue our study on what all these factors are in the future. The results of our experiments so far are these.
Reference
- Liang, Yixiong, et al. "Comparison detector for cervical cell/clumps detection in the limited data scenario." Neurocompu 499 ting 437 (2021): 195-205.
- Jiang, Hao, et al. "Deep Learning for Computational Cytology: A Survey." arXiv preprint arXiv:2202.05126 (2022).
Appendix:

Reviewer 2 Report
I don't see any major improvements after revision. Many of my previous points were not included in the revision.
Author Response
Response to Reviewer 2 Comments
Comments:
I don't see any major improvements after revision. Many of my previous points were not included in the revision.
Response to Comments:
We apologize for not letting you see the changes I made in the paper and for not clearly pointing out the changes, which was our problem. We have carefully reviewed and revised our paper , and have clearly identified the changes below.
Point 1: The motivation towards selecting the architecture is not clear.
Response 1:
We introduced our motivation for the paper in the abstract section. We are very sorry that we did not let you understand our motivation for the paper, this is our problem and we have revised the abstract section in the revised manuscript.
Regarding the motivation of our paper, I would like to introduce in detail. In the course of our survey on cervical cancer detection, we found that in recent years, with the success of deep learning in many fields, many researchers have used deep learning approaches to study automatic cervical cancer screening and diagnosis. Deep-learning-based Convolutional Neural Networks (CNN) models require large amounts of data for training, but large cervical cell datasets with annotations are difficult to obtain. Some studies have used transfer learning approaches to handle this problem. However, we found that in these studies, the detection task was performed on cervical cell/clumps using the same transfer learning approach as the classification task, i.e., the backbone network was initialized with an ImageNet pre-trained model. We propose to use the COCO pre-trained detection model for initialization, considering that the detection task and the classification task are two different tasks. To further improve the detection precision of the network model in the current dataset, based on transfer learning, we performed multi-scale training; fine-tuning strategy; analyzed the impact of different bounding box losses on the network model and selected the most suitable for the current situation: SmoothL1 loss; we analyzed the impact of the mean and std of different datasets on the performance of the current model,the best one is selected.
In order to more clearly express our motivation, we have modified the abstract section. First place: In line 17, the abstract section, we have changed the original “such studies used the same transfer learning method initialized by the ImageNet pre-trained model on the backbone network in two different types of tasks for the detection and classification of cervical cells” to the current “such studies used the same transfer learning method that is the backbone network initialization by the ImageNet pre-trained model in two different types of tasks, the detection and classification of cervical cell/clumps” Second place: In line 27-29, the abstract section, we added the content “We analyzed the effect of mean and std of different datasets on the performance of the model. It was demonstrated that the detection performance was optimal when using the mean and std of the cervical cell dataset used in the current study.”
Point 2: The paper lacks explainablity and mathematical validation towards the selection of the model and loss functions.
Response 2: Thank you very much for your suggestion. We have supplemented relevant content in our paper.
Considering the amount of data, we used transfer learning in this study and experimentally verified that the model performance can be greatly improved by using transfer learning in the current data set.
We have made relevant additions to the loss function in Section 2.4.: from line 222 to line 267. The additions are as follows:
The mathematical definitions of bounding loss functions are shown below.
- L1 loss:
x: The difference between the predicted value and the true value. L1 loss is not smooth at the zero point.
- L2 loss:
L2 loss has a large x value and correspondingly large derivatives at the beginning of training, which makes the initial training unstable.
- SmoothL1 loss:
SmoothL1 loss combines L1 loss and L2 loss, which uses L1 loss when x is large at the initial stage of training.
- IoU loss:
A: Prediction box; B: Ground truth
IoU loss function for bounding box prediction, which regresses the four bounds of a predicted box as a whole unit.
- GIoU loss:
C: The smallest enclosing convex object for A and B.
When two boxes intersect, GIoU takes into account not only the overlapping part but also other non-overlapping parts, which better reflects the overlap of the two boxes.
- DIoU loss:
Definitions of the symbols are in Table 2.
DIoU takes into account not only the overlap area between the prediction box and the ground truth but also the central point distance.
- CIoU loss:
CIoU loss takes into account the three geometric factors in bounding box regression: overlap area, central point distance, and aspect ratio.
In this paper, the effects of different bounding box loss functions on model performance are considered based on transfer learning, and the most suitable bounding box loss function, SmoothL1, is selected by comparing the experimental results. See Section3.5 for details.
Table 2. Definitions of the symbols used in the bounding box function
|
Symbol |
Explanation |
|
The difference between the predicted value and the true value |
|
|
Prediction box; |
|
|
Ground truth |
|
|
For and , find the smallest enclosing convex object |
|
|
|
Central point of |
|
Center point of |
|
|
Euclidean distance |
|
|
the diagonal length of the smallest enclosing box covering and . |
|
|
positive trade-off parameter |
|
|
Measure the consistency of aspect ratio |
|
|
The width of |
|
|
The height of |
|
|
The width of |
|
|
The height of |
Point 3: Comparision with state of the art segmentation models such as U-Net, LinkNet in place of FPN have not been discussed.
Response 3:
The dataset used in our study is dedicated to the detection task. Segmentation and detection are two different types of tasks, and the format of the dataset annotation for performing the segmentation task is different from that for performing the detection task. Our dataset is specially used to detect cervical cell/clumps. The annotation information to be learned includes: the location information and category information of the object in the image. Annotated examples are shown in Fig. 2-1, which shows 00146.img in the train set used in the experiment, See Figure 2-2 for the detailed annotation information of this picture. This dataset in our experiment has no annotation information for performing the segmentation task, so the segmentation task cannot be performed.
Figure 2-1. 00146.img with category annotations
Figure 2-2. Example of image annotation information
Suggestion:
Point 3: The authors are suggested to add better representation of the outputs provided by the proposed model.
Response 3:
Thank you very much for your suggestion, we have made the output representation of the model better and clearer.See Figure 5 of the revised manuscript for details. In order to make the sample images with category annotations better and clearer, we have adjusted Figure 4 as well. We have adjusted Figure 3 as well.
Figure 4. Sample images with category annotations
|
|
|
|
(a) |
(b) |
Figure 5. Detection results on the test image. (a) Left images are test images with ground truth; (b) right images are the predicted images.
Point 4: The authors are suggested to add an ablation study wherein the change in accuracy scores (mAP and AR) with respect to the number of data-samples are mentioned.
Response 4: We have carefully considered this proposal from the following aspects.
(1) The dataset used in this study has a small sample size for some categories, so it is not feasible to reduce some samples, which could easily result in missing categories.
(2) Some cervical cell images have multiple categories of ground truth, and it is difficult to reduce the sample size in the same proportion.
(3) Importantly, the purpose of our experiments is to demonstrate that transfer learning works well in the current dataset. In addition, from a deep learning perspective, as the amount of data decreases, the performance will decrease, so it is less meaningful to do experiments.
(4) Thank you very much for your suggestion, and we will discuss this issue in depth again if a suitable dataset becomes available in the future.
Point 5: The literature review is very trivial. The authors are encouraged to add the
paper:https://pubmed.ncbi.nlm.nih.gov/34328530/
https://www.sciencedirect.com/science/article/pii/S00104825220
07910?dgcid=rss_sd_all
Response 5:
We thank you for your suggestions and we have read the two papers carefully.
We cited the paper: https://www.sciencedirect.com/science/article/pii/S0010482522007910?dgcid =rss_sd_all in the Introduction of our paper and added it to the reference list. By reading the paper: https://pubmed.ncbi.nlm.nih.gov/34328530/", we found that the paper uses machine learning (ML) algorithms. Our study is based on deep learning algorithms, so the paper is not cited.
The reference number cited is "[11]", which we have cited to line 62(page 2) of the Introduction section.
We thank you again for your suggestions, which have been very helpful to our study.

Round 3
Reviewer 2 Report
Authors have revised the paper as per the comments.